

**Saturating response of photosynthesis to increasing leaf area index allows selective harvest of trees without affecting forest productivity**

**Olivier Bouriaud [1,2*], Ernst-Detlef Schulze [3], Konstantin Gregor [4], Issam Boukhris [5], Peter Högberg [6], Roland Irslinger [7], Phillip Papastefanou [3], Julia Pongratz [8,9], Anja Rammig [4], Riccardo Valentini [5], Christian Körner [10]**

1. Ștefan cel Mare University of Suceava, Str. Universității 13, 720229 Suceava, Romania. (obouriaud@usm.ro)

2. ENSG, IGN, Laboratoire d'Inventaire Forestier, 54000 Nancy, France.

3. Max Planck Institute for Biogeochemistry, Jena, Germany. (dschulze@bgc-jena.mpg.de, papa@bgc-jena.mpg.de)

4. Land Surface-Atmosphere Interactions, Technical University of Munich. (anja.rammig@tum.de, konstantin.gregor@tum.de)

5. University of Tuscia, Dept of Forest Environment and Resources, 01100 Viterbo, Italy. (rik@unitus.it, issamboukhris@gmail.com)

6. Department of Forest Ecology and Management, Swedish University of Agricultural Sciences, SE-901 83 Umeå, Sweden. (Peter.Hogberg@slu.se)

7. Hochschule für Forstwirtschaft Rottenburg, Schadenweilerhof, Rottenburg a.N., Germany (irslinger@gmx.de)

8. Ludwig-Maximilians-Universität München (DE) (julia.pongratz@lmu.de)

9. Max Planck Institute for Meteorology (Hamburg, DE)

10. University of Basel Department of Environmental Sciences Plant Ecology and Evolution, Schönbeintrasse 6. CH-4056 Basel. (ch.koerner@unibas.ch)

**Corresponding author:** Olivier Bouriaud, ORCID# 0000-0002-8046-466X, obouriaud@usm.ro

**This file includes:**
Main Text (4904 words)
Figures 1 to 5
Tables 1 to 2
Supplementary 1 to 3

**Key Points:**

- In temperate forests, net $CO_2$ uptake remains constant after partial harvesting.
- The relation between Gross primary production (GPP) and leaf area index (LAI) shows saturation above a threshold of 4-5 $m^2\ m^{-2}$.
- Harvest-related reduction of leaf area thus has little effects on the uptake if LAI remains above the threshold.



**Abstract**
Maintaining or increasing forest carbon sinks is considered essential to mitigate the rise of
atmospheric $CO_2$ concentrations. Harvesting trees is perceived as having negative
consequences on both the standing biomass stocks and the carbon sink strength. However,
harvesting needs to be examined from a forest stand canopy perspective since carbon
assimilation occurs in the canopy. Here we show that a threshold of leaf area exists beyond
which additional leaves do not contribute to $CO_2$ uptake. The associated biomass can be
harvested without affecting the forest carbon uptake. Based on eddy covariance
measurements we show that $CO_2$ uptake (GPP) and net ecosystem exchange (NEE) in
temperate forests are of similar magnitude in both unmanaged and sustainably managed
forests, in the order of 1500-1600 gC m$^{-2}$ y$^{-1}$ for GPP and 542 – 483 gC m$^{-2}$ y$^{-1}$ for NEE. A
threshold of about 4 m$^2$ m$^{-2}$ LAI (leaf area index) can be used as a threshold of sustainable
harvesting with regard to $CO_2$ uptake. Simulations based on the LPJ-GUESS model
reproduce the saturation of GPP and NEP and convergence on the LAI threshold range.
Accordingly, in managed forests, trees can be harvested while maintaining a high tree
biomass and carbon sink of the remaining stand. In this case, competition between neighbor
trees in unmanaged forests is replaced by harvest management and provision of wood
products. In unmanaged forests, competition for light, nutrient and water cause self-thinning,
thereby limiting the carbon sink strength.

**Introduction**
At times of increasing global change and a demand for wood to replace fossil fuel
products, it becomes of eminent importance to know if forest management and wood
harvest counteracts climate change mitigation. Following the EU definitions on storage and
uptake respectively (EU 2018), two major ways exist by which forests may contribute to the
efforts of climate mitigation: the storage of biomass on site within the forest ecosystem and
the storage of wood in products or their use for substitution of fossil-fuel or carbon-intensive
materials (Gregor et al., 2024). It is assumed that storage and C stocks can be sustained or
increased only by increasing the area of forests, or by stopping wood procurement from
forests (no management). However, halting management will probably have little effects on
the forest carbon sink and long-term stocks at landscape level, considering the environmental
risks associated with climate change that strongly increase the chances of stand collapse
(Roebroek et al., 2023). Furthermore, ageing forests have large biomass stocks, hence a large
C storage, but a very low growth translating into a very low C sink strength once they reach a
natural equilibrium. Forest stocks are thus finite on a given forested land area, with a possible
saturation already reached in European forests (Nabuurs et al. 2013) and this storage capacity
depends on the environmental conditions (Vetter et al., 2005). In contrast, managing forests
for products can be continued nearly endlessly if management is performed in a sustainable
way (Carlowitz, 1713; MCPFE, 1993). According to Pretzsch et al. (2023), self-thinning
losses could be equivalent to wood extraction by management. Luyssaert et al. 2011 also
show that management keeps forest stands close but below self-thinning, albeit at different





stand density and volume. Besides ensuring a sustained carbon sink, harvesting wood
products can substitute carbon-intensive materials and the energy use of wood residues and
end-of-life wood products can substitute energy from fossil fuels (Cowie et al., 2021; Schulze
et al., 2022). However, the provision of wood, even from selective cuttings, is considered as a
disturbance for the forest ecosystem, particularly for the carbon sink strength. A reduced
growth may in turn slow down the recuperation of the stocks after harvesting. Thus,
understanding the consequences of selective harvesting on the carbon balance and sink
strength of forests after disturbance is a key element to future projections on the role of
forests to climate change mitigation.

While harvesting is seen as a disturbance, forest productivity is not necessarily affected by
selective harvesting (including various forms of thinning) across a large range of cutting
intensities (Skovsgaard 2009), suggesting that the assimilation of carbon by forest stands is
not always reduced by harvesting (Amiro et al., 2010; Peters et al., 2013; Bond-Lamberty et
al., 2015; Noormets et al., 2015). The mechanisms involved in explaining the resilience of
productivity to management are based on the enhanced productivity of the remaining trees.
Reasons for this are, for example, improved light conditions, nutrient and water supply and
overall light use (Mund et al., 2010; Saunders et al., 2012; Sohn et al., 2016; del Campo et al.,
2022). Compensatory contribution of subcanopy individuals can locally also be observed
(Vesala et al., 2015). In previous studies several such factors and interaction pathways have
been identified (e.g., Noormets et al., 2015, Fig. 1) but canopy density, as quantified by leaf
area index (LAI, the cumulated area of leaves per ground square meter, expressed in $m^2 \ m^{-2}$)
was not taken into consideration despite its key role in $CO_2$ uptake.

Here, we introduce the link between photosynthesis and leaf area as a key element in this
regulation. We hypothesize that LAI is not only the link between the atmosphere and the
plant, but is also central to the response to management. LAI is indeed largely seen as a
driver of both water and carbon fluxes (Reich, 2012; del Campo et al., 2022). Given its high
nutrient demand the production of leaves also affect the nutrient cycle (Ollinger et al., 2008)
and is a potentially crucial driver of forests response to harvesting.

Harvesting inevitably results in a reduction of the amount of canopy leaves, best quantified
by LAI. It can be assumed that a reduction of LAI would lead to a decrease in productivity.
However, there are indications of a saturation of several canopy processes resulting in a non-
linear relation between leaf area index at stand level (Soimakallio et al., 2021) that make the
response of productivity to disturbances complex and difficult to predict (Glatthorn et al.,
2017; Stuart-Haëntjens et al., 2015). For principal reasons, a rise in LAI must have
diminishing returns in terms of light capture and $CO_2$ assimilation, given the exponential light
extinction with canopy depth, as described by Monsi and Saieki 1953 (see Hirose 2005).
Concerning canopy conductance, Schulze et al 1994 concluded to a saturation of around 3.5
$m^2 \ m^{-2}$. These elements suggest that productivity could have a non-linear response to
reductions of LAI and hence, to management while examined at stand level. Regardless of



the mechanisms, however, the effects appear beyond a yet unknown level of biomass
removal. A comparison across temperate forests beyond the site-level analyses is lacking.

The impact of harvest on the C cycle is clearly of major importance in the public debate. It is
thus necessary to determine the impact of harvesting on the fluxes of carbon in forests based
on experimental data over a large gradient, and to discuss the limits in the context of leaf area
reduction. In particular, the interactions between management and LAI, and their
consequences for the carbon sink strength need to be determined in order to examine the
consequences of wood harvesting on forests carbon sink strength. Here we intend to show
that sustainable management replaces natural competition by regulating leaf area without
affecting ecosystem fluxes in temperate forests. Based on observational data, literature and
modeling we want to identify mechanistic reasons for this presumption and explore the
possibilities of defining levels of sustainable partial cuttings from the perspective of carbon
fluxes, key to designing forest managements strategies able to maintain high biomass as well
as forest C uptake over multiple cutting cycles.
**Materials and methods**
***Observational flux data based on eddy covariance measurements on the FLUXNET sites.***
Overall FLUXNET represents 212 sites worldwide of eddy covariance. In order to measure
the impact of management over the carbon fluxes, we have compiled flux data from the 29
FLUXNET sites (https://fluxnet.org/data/fluxnet2015-dataset/) that comprise 19 managed and
10 unmanaged sites (unmanaged is used in the sense of "intact" forests of Roebroek et al.,
2023) with long-term measurements in temperate forests (**Supp. Table S1**). Unfortunately,
there is no site that covers unmanaged conifers. For each site we have compiled the forest
type, stand type, and the fluxes over their monitoring period. We completed these data with
estimations of the LAI during the period 2000-2020 and of the standing biomass.
Noticeably, selective harvesting took place on 11 of the managed sites during the period of
flux monitoring, several interventions being quite intensive (Supp. Table S3): for instance,
36% LAI removal in Fontainebleau site (FR), 30% removal in Bily Kriz site (CZ). Other
managed sites have experienced interventions prior to the monitoring but not necessarily
during the monitoring period, given the long periods of time separating interventions.
Furthermore, during the period of flux monitoring, forests experienced repeated events of
storm, drought and heat such as that of 2003, affecting ecosystem fluxes independent of
management.
Further, we have compiled LAI estimations for the analyses, for each of the FLUXNET sites.
LAI measurements, however, are not standard across sites, and field measurements are not
always available (5 sites had no field measurements). In this situation remote-sensed
estimations were used instead based on the MCD15A3H version 6.1 MODIS data level 4 (see
**Supplementary Table S1**, with references for each estimation).

The eddy covariance method does not actually measure the fluxes but instead measures
atmospheric $CO_2$ concentrations and wind speed which are converted into fluxes, i.e., the net



ecosystem exchange (NEE), with different levels of uncertainty. Fluxes data were filtered
based on USTAR threshold levels according to Pastorello et al. (2020) to account for errors of
measurement at low levels of turbulence. Errors have been estimated using bootstrapping 200
times with different friction velocity values.
The fluxes of carbon exchanged between the forest ecosystem and the atmosphere are
generally divided into components that are physiologically meaningful: the gross primary
production (GPP) corresponds to the photosynthesis of plants, and the ecosystem respiration
(Reco) releasing $CO_2$. Reco consists of plant respiration (so-called autotrophic respiration)
and respiration by heterotrophic organisms (so-called heterotrophic respiration). The NEE
can be estimated by eddy covariance, partitioning into the other elementary fluxes follows
data-driven models (Valentini et al., 2002).
We compared the mean fluxes during the period of time available of managed and
unmanaged sites. For testing the significance of differences in NEE we used the Wilcoxon
rank test because data were not distributed normally. GPP and Reco have a distribution that
does not differ significantly from a normal distribution. The Mann-Whitney test has been
implemented to compare managed versus unmanaged sites which works with unequal sample
sizes. For GPP and Reco, their distributions being normal, but their variances unequal, the
Welch t-test was used instead. Subsequently, two-way analysis of variance for unbalanced
designs was performed on the data to check if the interaction between the management and
the number of observations by FLUXNET site has a significant effect on GPP, Reco, and
NEE.
The relationship between GPP and LAI for the FLUXNET observational site was represented
as a nonlinear asymptotical model. The fitting was based on the nonlinear fit function *nls* (nls
standing for nonlinear least square) in R. The pseudo-$R^2$ represents the proportion of variance
that was explained by the model, in lieu of the $R^2$ which assumptions cannot be completely
satisfied with nonlinear models (Schabenberger and Pierce 2002). It was computed as
*pseudo-$R^2$* $= 1 - (\mathrm{var}(y_{fit})/\mathrm{var}(y))$, where $\mathrm{var}(y_{fit})$ is the variance of the predicted value (GPP
here), while $\mathrm{var}(y)$ is the variance of the variable (GPP) within the dataset.
***Harvesting and carbon fluxes***
Harvesting takes many forms in forest management and can have different intensities.
Harvesting is defined in a general way as the removal of wood by tree cuttings of any kind,
thus including tending, thinning (targeting either dominant or sub-dominant trees) and
selective cuttings from either status. While short- and medium-term effects of selective
harvesting are being considered, this study will not cover the comparison of forest products
with other bioenergy sources (product and energy substitution). In the following, clear-
cutting, or final felling of a rotation, are treated separately from selective cuttings as they
need an assessment at landscape or management unit-scale. The measurement of carbon
fluxes using the EC method is limited to a plot-scale, with a footprint commonly of about 1
km$^2$. Throughout this study, harvesting refers to practices of selective harvesting at low to
moderate intensity as common in temperate forests. For example, removal of harvest



residuals is widely seen as negative because of the nutrient and soil carbon depletion it causes
(Achat et al., 2015, Mayer et al., 2020).


***Modelling analysis of the impact of an increasing LAI gradient on CO$_2$ fluxes exchanged,***
***using the process-based model.***
To investigate the impact of LAI on GPP, we used the dynamic global vegetation model LPJ-
GUESS v4.1.1 (Smith et al., 2014, Nord, 2021) to simulate the main carbon fluxes (GPP,
Reco and NEP) on all the eddy-covariance sites used in the study. The ability of LPJ-GUESS
to estimate LAI and GPP values worldwide has been proven in numerous studies (e.g., Vella
et al. 2023 and Ito et al. 2017, see also Fig. SF2). Therefore, the model is well suited for the
analyses. LPJ-GUESS simulates detailed vegetation structure (including cohorts of various
ages) based on mechanistic modeling of ecosystem processes including photosynthesis,
establishment, growth, allocation, competition, water and nutrient limitation, and mortality of
plant functional types (PFTs). The latter are represented by parameters defining plant
characteristics such as bioclimatic limits, growth form, or shade-tolerance.
In the model, at the end of each year, cumulative net primary productivity is distributed
among the leaf, root, sapwood and heartwood compartments of a plant, based on allometric
equations and allocation routines per year (Smith et al., 2014). LAI is calculated as the
product of the carbon mass of the leaves times the specific leaf area, the specific leaf area
being a PFT parameter. LAI is computed proportionally to the phenology fraction of the
PFTs, that is, the fraction of potential leaf cover. The phenology of a PFT can be raingreen,
summergreen or evergreen. LAI is also influenced by the phenology: depending on the
environmental conditions, the phenology fraction can depend on growing degree days and
drought stress related model states. The amount of light taken up by the canopy, and thus
contributing to carbon allocation, is governed by LAI, based on the Lambert-beer law
(Prentice et al, 1993). The model outputs stand level LAI, taking into account the number of
trees per area and the crown areas of the various cohorts.

For the LAI analysis, we ran LPJ-GUESS until 2015 using daily climate data from the
FLUXNET2015 sites, i.e., precipitation, temperature, and shortwave radiation. For each site,
we prescribed the forest type as described in Table S2. We used 1000 years for the spinup
period (to bring soil pools close to equilibrium) by detrending and recycling the first 10 years
of each site's climate data. CO$_2$ concentrations were taken from (Büchner and Reyer, 2022).
We used the default global parametrization of LPJ-GUESS with global PFTs, without any
form of management.
Stochastic disturbance intervals were kept at default values while fire was not simulated.

**Results**

***Saturated response of fluxes to LAI***



Regular management actions were performed in most of the managed sites during the
monitoring period with removals as high as 30% of the stems for some sites during the
monitoring period (**Sup. Table 3**). Managed sites are mostly age-selection (forests stands
composed of trees of similar age, obtained from harvesting trees at a prescribed age) and
plantations. In the whole flux network, there is only one pair of managed and unmanaged
sites: DE-Hai (Hainich, unmanaged) and DE-Lnf (Leinefelde, managed) representing *Fagus*
stands with similar stand densities or basal area.

The data from the FLUXNET sites show a response of GPP to LAI only for LAI values less
than ~4 $m^2$ $m^{-2}$ (**Fig. 1**) but GPP does not increase at higher LAI.  It is interesting to note that
most managed forests operate near the range of saturating LAI, despite harvesting. Likewise,
the data shows a saturation of GPP even in managed sites, with values reaching a plateau in
the order of 1770 gC $m^{-2}$ $year^{-1}$ at LAI values as low as 4 $m^2$ $m^{-2}$. Based on the GPP-LAI
regression, 95% of GPP (1680 gC $m^{-2}$ $year^{-1}$) is reached at LAI of 4.5 $m^2$ $m^{-2}$. The exact
location of the LAI saturation point can only be approximated given the uncertainty in both
LAI and C flux data. The site at Parco Ticino Forest (Italy) has been fertilized. It indicates the
importance of nutrition in forest ecosystems as a GPP value above 1800 gC $m^2$ $y^{-1}$ was
reached at low LAI (< 2 $m^2$ $m^{-2}$). However, even with fertilization, the fluxes and LAI values
remain in the range of other sites. Reco had a smaller overall variability than GPP (1082 ±
151 gC $m^2$ $y^{-1}$) and showed no response to LAI. Likewise, there was no response to forest
types. The net ecosystem exchange (the balance between photosynthesis and respiration, GPP
– Reco = NEP) did not show any significant response to LAI, with values largely scattered
around the mean (343 ± 151 gC $m^{-2}$ $year^{-1}$).
The data represent a mixture of remotely-sensed and field-based LAI for different forest
types. Given the large variability among sites, differences in fluxes for managed and
unmanaged forests in **Figure 1** are not significant (**Table 1**).
It is notable that, although not significant, LAI tended to be higher under management (4.74 ±
1.33 for managed sites versus 4.40 ± 0.82 $m^2$ $m^{-2}$ for unmanaged sites, n.s.), despite the
removal of parts of the canopy due to management in the past (**Fig. 2**). LAI was indeed
strongly reduced during the monitoring period by thinnings ranging from 26 to 36% in four
of the managed sites (**Sup Table 3**). For instance, the low (3.6 $m^2$ $m^{-2}$) LAI value at site CS-
BK1 (*Picea abies* L.) reflects the 26% removal that occurred at the end of the monitoring
period. The dynamic of LAI on the sites show that the reduction of the LAI by harvesting is
limited to a few years following the harvesting (Sup Fig. 1).

***Responses of fluxes to sustainable harvesting: empirical evidence from eddy covariance***
The FLUXNET associated site data showed that past and current management has little
influence on the aboveground biomass and LAI of the sites (**Fig. 2**).  Highest biomass was
reached with the old-growth *Eucalyptus regnans* site in Australia (Wallaby Creek site, with
36.106 g dry matter $m^{-2}$). Unfortunately, there is no managed site of *E. regnans* for
comparison. Otherwise, the range of values is very similar among managed and unmanaged
sites.



The comparison of the fluxes reveals that the net ecosystem exchange (the balance between
photosynthesis and respiration) was not significantly different in managed and unmanaged
sites (-542 ± 219 gC m$^{-2}$ year$^{-1}$ for managed sites against -483 ± 306 gC m$^{-2}$ year$^{-1}$, mean ±
sd for unmanaged sites) over an observation period of more than a decade (**Table 2**).
Management was not a significant effect for GPP or NEP. As shown in **Fig. 3**, Reco and GPP
tended to be higher in managed sites (Reco: 1213 ± 121 gC m$^{-2}$ year$^{-1}$ in managed sites
versus 1079 ± 98 in unmanaged sites; GPP: 1715 ± 192 gC m$^{-2}$ year$^{-1}$ in managed sites
versus 1489 ± 183 gC m$^{-2}$ year$^{-1}$). The paired DE-Hai and DE-Lnf unmanaged sites had very
similar values of both GPP (1709 gC m$^{-2}$ year$^{-1}$ in the managed site DE-Lnf vs. 1653 gC m$^{-2}$
year$^{-1}$) and NEP (1189 vs 1155 gC m$^{-2}$ year$^{-1}$). We investigated whether the forest type had
any influence on the LAI or the fluxes, since conifers tend to have higher LAI values with
few exceptions. A linear model was fitted to the data and showed no significant influence of
management or forest type (**Table 2**). Interactions between forest type and management were
not significant either.
*Process based model simulations: sensitivity to LAI*
We applied the LPJ-GUESS process-based dynamic vegetation-terrestrial ecosystem model to
further investigate the relationship between LAI and GPP, Reco and NEP, on each of the
FLUXNET sites. Within a given site, GPP increased with LAI, near linearly for LAI < 3 m$^2$
m$^{-2}$, showing a clear inflection around this value (**Fig. 4**). Saturation is visible at high LAI
sites at around 4.5 m$^2$ m$^{-2}$ and above. Reco followed a very similar pattern, albeit starting at
higher values for very low LAI level and having a smaller increase with LAI than GPP. GPP
and Reco curves cross each other at different LAI values (between 1 and 3 m$^2$ m$^{-2}$) depending
on the sites, at which point NEP becomes positive but shows a strong saturation after with no
response at all to LAI. Thus, NEP becomes positive (forest acts as a sink) for LAI in excess
of 3 m$^2$ m$^{-2}$ but, beyond 4 m$^2$ m$^{-2}$, increases in LAI do not result in increases in NEP.
**Discussion**
With the introduction of the eddy covariance method, long time series of carbon fluxes
became available over a variety of biomes, with most monitoring sites being under regular
forest management (Franz et al., 2018). Based on these time series, our synthesis showed here
that GPP and NEE remain largely unaffected by partial harvesting, as also reported by site-
level analyses for several forest types and species (Granier et al., 2008; Launianen et al.,
2022; Lindroth et al., 2018; Pilegaard et al., 2011; Peichl et al., 2022; Vesala et al., 2005).
Vesala et al., 2005 observed no visible effects of thinnings on the NEE despite the reduction
of LAI from 8 to 6 m$^2$ m$^{-2}$ in a Scots pine stand. Granier et al. (2008) reported for Fagus
stands no decrease in either NEE or GPP despite the thinning that decreased LAI from 7.4 to
4.8 m$^2$ m$^{-2}$. These results are in agreement with Herbst et al. (2015) and are confirmed by the
global database of Luyssaert et al. (2007) which shows that managed forests globally
achieved similar, or even larger GPP, than unmanaged forests. A contribution to the lack of
response of eddy covariance fluxes to harvest could be caused by the geographic
displacement of the respiration (typically occurring outside forest when the wood products



are burned) and the discrepancy between the location where harvest occurs and the eddy
covariance's footprint (Schulze et al., 2022).
The harvest effect on LAI appears to be short-term in temperate forests (del Campo et al.,
2022) as also suggested by the available LAI time-series of the sites studied here (**Supp. Fig.**
**SF1**). For instance, according to Granier et al. (2008) LAI in Fagus stands was restored to its
pre-thinning level within two years. Disturbances, particularly stand-replacing disturbances
such as windthrow, fire or clear-cuts have a different dimension and need to be evaluated at
landscape scale. Our study deals with thinning operations where the main canopy is reduced
but not removed, keeping LAI beyond or near to its saturation threshold. This also justifies
the choice of focusing on temperate forests where the lower species richness and age ranges
may slow the recovery of carbon uptake to catastrophic events, in contrast to tropical forests
(Brando et al., 2019). For boreal forests, the IBFRA-Report (Högberg et al., 2021) shows that
biomass increased significantly over the past decades only in intensively managed
landscapes, but not in less intensively managed forest landscapes (i.e., landscapes with a high
proportion of unmanaged forests). In the latter, large-scale disturbances such as wildfires
caused losses of biomass and prevented a build-up of forest carbon stocks. In comparison, the
biomass gain in non-managed temperate forests is very small (Roerbroek et al., 2023).
Roerbroek et al. (2023) indeed suggests that betting on increasing the forests stocks is not
only risky, given the increases in weather extremes, but loses the societal benefit of wood
products as well as the potential to store a portion of the C over longer term.
We propose that most of the decoupling between selective harvesting and $CO_2$ fluxes is
mediated by the intrinsically nonlinear response of the dominant processes to LAI with a
saturation point reached at 4-5 $m^2$ $m^{-2}$. This nonlinear response, particularly the existence of a
saturation point, is related to the existence of a fraction of the canopy leaf area not necessary
for productivity but serving other functions such as competition, or redundancy in case of
competition. In forest management it is known that about a third of the green foliaged tree
crown can be pruned to improve stem quality without affecting growth (Burschel and Huss
2003). Diffuse light can penetrate deeper into the canopy and reach lower levels of leaves,
but the gain in photosynthesis may not counterbalance the cost of producing and maintaining
saturated canopies. The carbon balance of a living branch may be close to the light
compensation point of photosynthesis and respiration (Schulze 1970), with a photosynthesis
activity just at the level needed to keep a shaded branch alive. Similarly, in the simulations of
the model LPJ-GUESS, small trees with low LAI operate at a higher level of light extinction
due to shadowing by bigger trees, which leads to very low GPP as no direct sunlight can
reach any leaves (Fig. 4). Shadowing also leads to a reduction in Reco, however a minimum
maintenance respiration of the leaves is always needed to sustain functioning of the leaves.
While shade tolerance varies among species (Ameztegui et al., 2016), as reflected by
different maximum LAI values (Valladares and Niinemets 2008), the threshold for light
compensation is probably very similar across forest types or species despite variations in the
canopy structure. This suggests that increasing LAI beyond a demand-driven threshold has
other functions, for instance a competitive function with neighboring trees (Pretzsch and
Schütze 2009, Jucker et al., 2014) not only for light but also for nutrients (e.g., in a pre-



emption strategy, Craine and Dybzinski 2013), as a buffer against disturbance (e.g.,
herbivory) and a pool of nutrient reserves, ready for rapid re-allocation in case of sudden
demand (Körner 2009). Anten (2005) shows that canopy photosynthesis models predict LAI
values greater than optimal values for photosynthesis and quote theoretical studies that
conclude to a LAI always exceeding the physiologically optimal value for competitive
purposes. Avoiding a neighbor increases the resources of water and nutrients for the
dominant tree. This surplus fraction is temporarily diminished by selective harvesting,
explaining the lack of response of the main C fluxes at canopy level across a wide range of
LAI. Accordingly, a moderated management can be seen as a substitution of self-thinning
when forest stands are kept close but below self-thinning density levels (Luyssaert et al.,
392    2011).


These non-linear relations of a variety of processes with LAI caused by a saturation of GPP
and NEE at values around 4-5 $m^2$ $m^{-2}$ (see ex. Asner et al., 2003; Hirose 2005) have long
been known, although not previously related to the resilience to selective harvesting. This
includes ecosystem respiration: according to Zhao et al. (2021), at high LAI, respiration -
particularly heterotrophic respiration- increases faster than GPP, which results in a reduction
of NPP for values larger than 5.6 $m^2$ $m^{-2}$. In our analysis, the model did not go so far as to
project a negative impact of LAI on NEP, but the high cost of producing and maintaining
leaves and particularly shade leaves (Niinemets 2010), largely suggests this. A similar result
was obtained using the model CASTANEA which reproduced the nonlinear responses of
fluxes to LAI (Davi et al., 2006). In contrast, field measurements based on leaf collection,
hemispherical photographs or light transmission through plants, frequently report values in
excess of 5 $m^2$ $m^{-2}$ (e.g., **Figure 3**) and even over 10 $m^2$ $m^{-2}$ in shade-tolerant species
(Schulze et al., 1994; Asner et al., 2003; Law et al., 2001; Iio and Ito, 2014). Out of the 29
sites we studied here (Fig. 1), 16 display LAI values in excess of 4.5 $m^2$ $m^{-2}$.
The lack of scaling between forest biomass and plant respiration (Piao et al., 2010) reflects
the fact that the mass of live tissues -that is, of respiring tissues- is much smaller than that of
total biomass, basically scaling to the parenchyma fraction in sapwood volume and small
branches only (Thurner et al., 2019). The disturbance-related increase in soil respiration, for
instance promoted by a short-term increase in root mortality (Raich and Nadelhoffer 1989),
could be comparable in magnitude to the reduction in plant respiration due to the amount of
sapwood harvested and the reduced influx of fresh litter (Davidson et al., 2002), and explain
the invariance of Reco. Surveying or modelling respiration has proved to be particularly
difficult (Phillips et al., 2017, Ciais et al., 2021) and results in uncertainties, which also
impact confidence in GPP estimates that could hide some effects. The lack of response of
Reco to LAI needs further investigations.

Unfortunately, the Hainich/Leinefelde *Fagus* sites are the only paired sites of managed versus
unmanaged sites within the flux network. The global eddy-flux network was indeed strongly
focused on climate as a main driver of fluxes, rather than management. The management
gradient represented by these sites is thus not complete, for instance the intensity and types of
management actions are not controlled. Although the unmanaged conifer sites are currently





not monitored, the NEP values for unmanaged conifer stands reported in synthesis studies
(Luyssaert et al., 2007) do not suggest that unmanaged conifer stands would behave
differently and have higher a NEP than managed ones. We nevertheless highlight the
potential of such paired studies and hope that research on management will be more
integrated in the future to improve our understanding of its short, medium and long-term
impact on the carbon balance of forests. We also underline the lack of common and frequent
reporting on the aboveground biomass and annual LAI on the FLUXNET sites, on harvested
volumes whenever management interventions occur. Annual measurements of LAI and
repeated study after disturbance should be considered. These critical data would strongly help
measure the impact of management on the carbon cycle.

## Conclusions

• Based on observational and modeling evidence, it appears that LAI regularly exceeds
levels required to sustain carbon assimilation in naturally growing forest ecosystems.
• Above its saturation value of ~ 4 $m^2$ $m^{-2}$, additional increases in LAI are not linked to
increased productivity, but may contribute to other functions selected in evolution,
such as competition with adjacent trees, resource storage and buffering against
herbivory.
• We can explain the lack of impact of harvesting on the $CO_2$ uptake by the existence of
non-linear processes governed that saturate around LAI values of 4 $m^2$ $m^{-2}$.
• Selective harvesting does not reduce the forest carbon sink strength when LAI is
maintained beyond its threshold.
• This threshold can be used to define sustainable metrics for sustainable harvesting, as
those that do not impact the carbon sink strength of the forest stand.
• Harmonized and periodic measurements of the forest carbon stock and LAI, and of
harvesting impacts on these, should be promoted at flux sites.

**Author Contributions**: Conceptualization, O.B., E.D.S. and C.K.; methodology, O.B. and
E.D.S.; writing original draft preparation O.B. and E.D.S. All authors contributed to the
writing, and reviewed the manuscript.
**Competing Interest Statement:** At least one of the (co-)authors is a member of the editorial board of
Biogeosciences.

## Acknowledgements

This work was supported by a grant of the Ministry of Research, Innovation and Digitization,
CNCS- UEFISCDI, project number PN-III-P4-PCE-2021-1677, within PNCDI III. KG
acknowledges funding by the Bavarian State Ministry of Science and the Arts in the context
of the Bavarian Climate Research Network (bayklif) through its BLIZ project (Grant No.
7831-26625- 2017, \url{www.bayklif-bliz.de}). RV and IB are supported by AGRITECH —
PNRR (Italian National Plan of Recovery and Resilience), identification code CN00000022



WP 4.3.3. Authors are very grateful to Susan Trumbore for her comments and suggestions on
the manuscript.

**Open research**
The data presented and analyzed in this study are available directly from the supplementary
information files, in tables S1 to S3. These tables also contain references to data sources.
Figures were made with R version 4.2 (R Core Team, 2021) (https://www.R-project.org/).

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





**Figures and Tables**

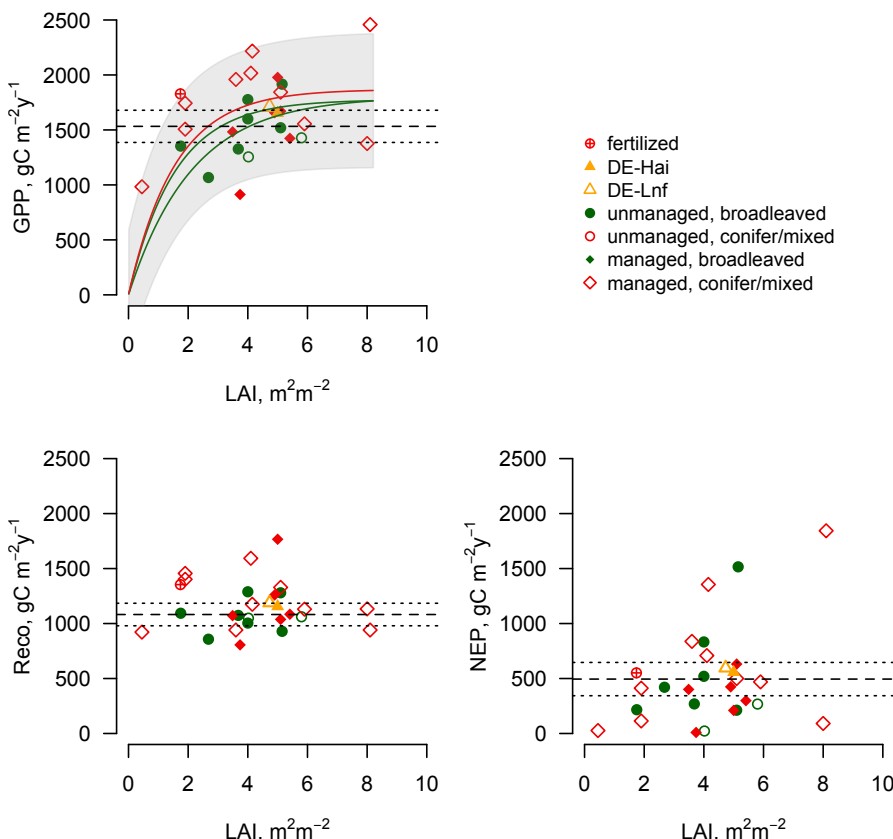


**Figure 1**. Relation between the GPP, the Reco, the NEP (= -NEE) and the LAI on the eddy
covariance sites (FLUXNET sites, see Supp. Table S1,2) of both managed and unmanaged
temperate forests per stand types.
The dashed lines represent mean and confidence interval of the GPP and NEP across all sites.
Curves show the fits for broadleaves (green), conifers and mixed forests (red), and all sites
together (black). The gray band represents the confidence interval of the regression on all
sites. The fertilized site is identified (Parco Ticino), along with the couple DE-Hai
(unmanaged) and DE-Lnf (managed). The exponential models illustrate the tendencies (Tab.
1), ±10% confidence intervals are displayed in gray.





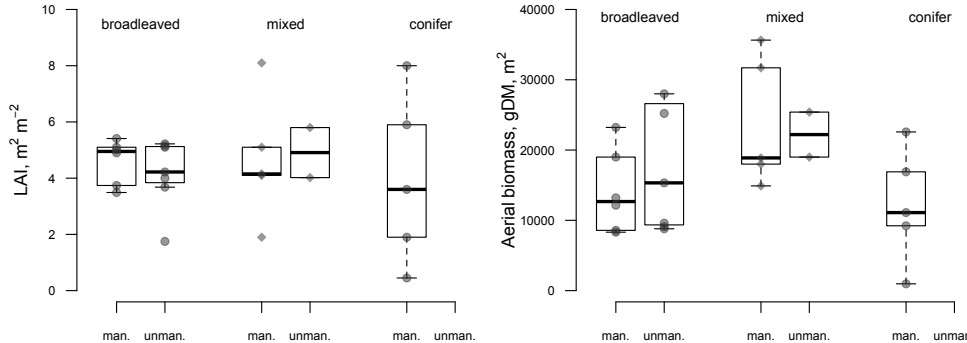


**Figure 2**. Comparison of the LAI and aboveground biomass values for the managed and
unmanaged sites, depending on the forest type. The site AU-Wac (Australia, natural
*Eucalyptus regnans*) is an extreme value due to low decomposition (Supp. Fig. 2) and was
not included in the biomass comparison.





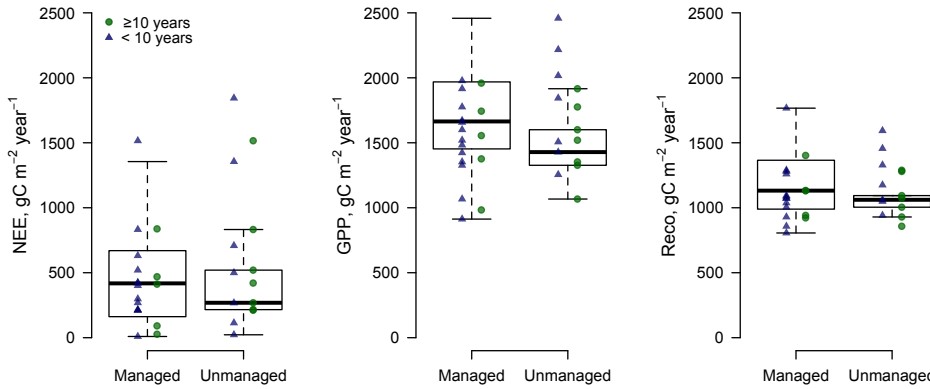

**Figure 3**. Comparison of the flux data from managed and unmanaged FLUXNET sites.
Dots represent the site-level mean values over the monitoring period.

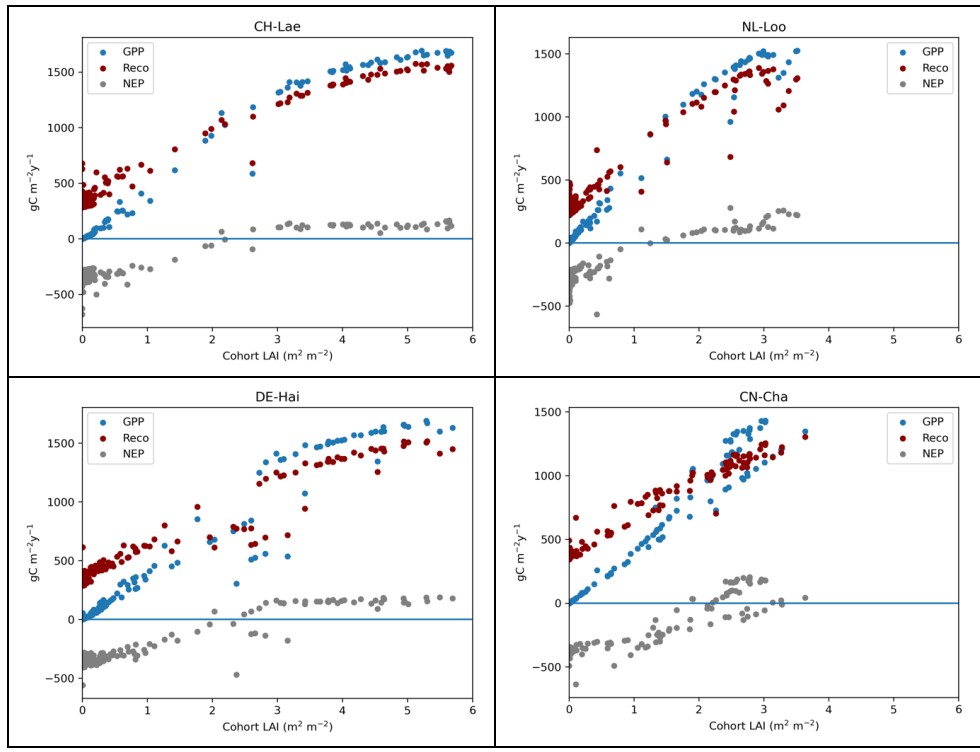

**Figure 4.** Variations of GPP, NEP and Reco along a gradient of LAI as modelled using LPJ-GUESS shown for 4 sites with contrasted maximum LAI and forest types: CH-Lae for mixed forest type with high LAI, NL-Loo for conifers with low LAI, DE-Hai broadleaved with high LAI and CN-Cha with low LAI broadleaved. Each dot represents the fluxes of a particular tree cohort simulated at a given site.

The model runs reveal that LAI in excess of ~ 4 $m^2$ $m^{-2}$ does not promote GPP or NEP. NEP becomes positive (forest acts as a sink) for LAI in excess of 3 $m^2$ $m^{-2}$ but, beyond 4 $m^2$ $m^{-2}$, increases in LAI do not result in increases in NEP.



**Table 1**. Effect of management type over the fluxes monitored on eddy correlation sites of
temperate northern-hemisphere (N = 29 FLUXNET sites, of which 18 managed and 10
unmanaged, after the exclusion of the Parco Ticino site (IT) of fertilized Populus), and fit
statistics of the nonlinear asymptotical models. Management is tested as a two-levels fixed
factor (managed/unmanaged) taken as Wilcoxon rank test for NEE, Welch t-test for GPP,
Reco and LAI. Pseudo-$R^2$ values were estimated from modeled and observed values (see
Methods section).

| *Flux* | Welch / t-test | *p-value* |
|---|---|---|
| NEE | W = 83 | 0.7595 |
| GPP | t = 1.745 | 0.0929 |
| Reco | t = 1. 711 | 0.0991 |
| *GPP ~ a\*(1 - exp(c\*LAI)), pseudo-$R^2$ = 0.517* | | |
| *Estimate (std error)* | *t value* | *Pr(>\|t\|)* |
| a = 996.798 (116.443) | 15.242 | 5.99e-16 |
| c = -0.184 (0.161) | -4.011 | 0.000354 |
| *NEE ~ a\*(b -exp(c\*LAI)), pseudo-$R^2$ = 0.935* | | |
| *Estimate (std error)* | *t value* | *Pr(>\|t\|)* |
| a = 648.998 (15180.454) | 0.043 | 0.966 |
| b = 1.199 (4.684) | 0.043 | 0.966 |
| c = -1.091 (51.191) | -0.79 | 0.938 |





**Table 2**. Estimation of the effect of management and forest type on the LAI or on the fluxes.
Interactions (management x type) were tested and not found significant, and are therefore not
presented here.

|  | Estimate | std. error | t value | Pr(>|t|) |
|---|---|---|---|---|
| LAI_mix ~ Management + type, $F_{(3, 25)} = 0.3592$, $p = 0.7829$ | | | | |
| Intercept | 4.233 | 0.789 | 5.358 | 1.48e-05*** |
| Management | 0.064 | 1.029 | 0.062 | 0.951 |
| Conifer | 1.209 | 1.258 | 0.961 | 0.346 |
| Mixed | 0.488 | 1.109 | 0.440 | 0.664 |
