# Peer review of "Saturating response of photosynthesis to increasing leaf area index allows selective harvest of trees without affecting forest productivity"

_EGUsphere, 2024_

## Community Comment (CC1)

**A comment on: Saturating response of photosynthesis to increasing leaf area index allows selective harvest of trees without affecting forest productivity**

This contribution is an important one, in that it provides a functional insight into a long-attested empirical demographic relationship relating plant/tree population density and yield. In spite of its robustness, this fundament of yield science and silviculture (being able to extract trees from a forest without sacrificing total yield, and increase growth of remaining trees in a process that fastens the acquisition of size, thus preserving its long-term sustainability) remains poorly perceived by ecologists and the general public, causing ill-posed debates in forest sustainability. For this reason, being able to support it by functional process-based grounding is major. This is an originality of the contribution.

That said, we also wish to share some matter of concern regarding specific aspects of the contribution, in order to give it the broadened perspective it deserves.

**1. Constant final yield**

One first aspect is the apparent disconnection there is, between this "modern" approach relating C fluxes and NEP from the EC methodology, and the bunch of historical work performed in forestry science to relate yield (ANPP) to stand density, also in agronomy science. The same saturation has been described as "law of final constant yield", "Langsaeter's plateau", the "thinning response hypothesis". Major texts include Yoda et al 1963, Assmann 1970, Pretzsch 2009, but also and more recently the nice synthesis by Weiner and Freckleton 2010 (named "constant final yield").

➔ Reference to these texts and concepts would allow better connect these findings to a long-established reality.

This also raises an issue. The following conclusive statement: *This threshold can be used to define sustainable metrics for sustainable harvesting, as those that do not impact the carbon sink strength of the forest stand* questions how it can be achieved. When this plateau was established in terms of ANPP = f (stand density), at least direct targets could be formulated for the prescription of silvicultural regimes. Yet, how may a forest manager easily pilot a LAI target?

➔ Here, I wonder whether plot density data are available, and to what extent the correlation between LAI and stand density (say BA, or N) would allow capture/confirm the alleged LAI threshold invariant identified, and may pave the way toward these metrics. In would rather incline toward acknowledging the merit of this transposition of initial stand density / yield relationship to LAI / photosynthetic C fixation to ground the previous relationships, and deliver the message that forestry science is justified to do so. Perhaps the aspect may be expanded both in the introduction and discussion.

**2. Tree species diversity and species traits**

As written, the paper gives an impression that finding the threshold is so motivating that the issue of variability is a bit discarded. At L277-279, it is even suggested that cross-species variation may be minor, but no reference is suggested for this aspect. By contrast, I would be of the primary opinion that the shade-tolerance of tree species may influence that potential threshold, light-demanding species meeting an earlier LAI threshold than others. This would be consistent with the notion of clear-forest silviculture as envisioned in the 19th century for the management of light-demanding species (e. g. pines, oaks).

➔ May it be useful to discuss the invariance of this LAI threshold in view of species traits? How far can we go? What may be the implications in terms of experimental designs based on current tower flux facilities? Is it an issue for a near future, or currently out of reach? This discussion would be welcome in the discussion.

Further, may the design structure of EC sites allow to explore, at least a bit, the issue, e. g. by ranging sites according to some community weighted mean of the shade-tolerance trait (TRY) and see whether it correlates to some parameter related to the fit of the non-linear responses to LAI? I have this impression that a prescriptive unique threshold value should be avoided.

➔ Is it possible, or prohibitive, to explore a quick statistical relationship between the position of the threshold and a community weighted-mean of shade tolerance?

**3. The role of excess LAI**

In the discussion, the adaptive significance of "excess LAI" is discussed, which suggests that excess LAI may serve the purpose of better tree resilience to abiotic or biotic disturbances. If so, while saturation is good news for the C storage, it is not necessarily the same for other functions related to resilience, in the present environment. Since it is indicated that some monitoring plots experience heat/drought or other disturbances over the period, may this be tested, or is it out of reach?

➔ I am uncertain about whether this test may be easy. At least, a better balance of the corresponding paragraph, and the trade-off it may induce in a silvicultural approach, is welcome.

**4. Methodological aspects**

In general, it is quite hard to grasp the monitoring period under study, precisely. And also to compare it to management events, in order to perhaps question the ranking of managed/unmanaged plots. It may seem worrisome that only 11/19 plots classified as "managed" did not show any management event over the monitoring period in view of the rapidity of LAI reconstitution in general.

➔ May this have contributed to dampen the difference between managed / non-managed forests? Shouldn't this be discussed a bit (and if possible, tested? And if untested, what would be the perspectives ?)

It was also very difficult to me to understand whether GPP/NEP data come from field plot measurement, or from LPJ-guess, and also what specific role does the model indeed play.

➔ I have this impression that the modelling objective may be specified more clearly in the dedicated section, and early in the introduction. E. g., is it because it is feared that the integration scale difference between plot and EC towers may have a role?

**5. Local detailed comments**

Line 78-79: Bontemps 2021 (Plos) does not suggest any saturation of the carbon stock in European forests

Line 95-98: the saturation of forest productivity with stand density has been a cornerstone of silviculture since the elaboration of the scientific principles of forest management. It is a bit frustrating it is not mentioned a bit more in-depth, a law known as the thinning response hypothesis, Langsaetter's law, and more generally in agricultural yield science, as the law of final constant yield.

In particular, *Saturating response of photosynthesis to increasing leaf area index allows selective harvest of trees without affecting forest productivity* may be rephrased as *Saturating response of forest ANPP to increasing stand density allows selective harvest of trees without affecting forest productivity.*

Line 122: why it would have impact on CO2 assimilation is less clear.

Line 128: and certainly also, across tree species, given that their sensibility to light (shade-tolerance) may have a contrasted effect on a unique LAI threshold.

Line 133-135: *In particular, the interactions between management and LAI, and their consequences for the carbon sink strength need to be determined in order to examine the consequences of wood harvesting on forests carbon sink strength.*

+ line 138-140: *explore the possibilities of defining levels of sustainable partial cuttings from the perspective of carbon 140 fluxes, key to designing forest managements strategies*

But how in practice, can we transfer a target N trees / ha under management into a LAI reduction / ha, especially since selective thinning can be deployed for different strategies (thinning from above / below)?

Line 154-155: what is the monitoring period exactly? Is it 2000-2020 as suggested by LAI measurements? If so, one may wonder about the significance of being classified as managed forests for those 8 study sites out of 19 that were matter of no harvesting over the period.

Line 164-166: were these MODIS-based estimates compared against measurements for those sites where site measurements for LAI do exist?

Line 168-173: in addition, may it be possible to specify a bit across which spatial range do the flux measurements integrate and to what extent there are representative of fluxes within the perimeter of the study sites?

Line 183-185: then if GPP and Reco are gaussian, why the difference = NEE would not be so? Are their discrepancies to be expected in this budget?

L 203: but also suppressed trees, with varying LAI

Line 209-210: then how can it be that measured fluxes are representative of forest plot dynamic?

Line 218: on monitoring plots, wasn't it possible to more directly measure tree-driven GPP, or at least compare?

Line 145-246: at this point, the issue of variability across tree species turns an issue. What are the species covered in the 30 monitoring plots? How do they match the PFT of LPJ-guess? To what extent there is a risk to remove this variability?

L253-254: can it be specified whether the management operations described on the monitoring plots were introduced into LPJ-guess? Or else?

L262-263: I would say this (should) has(ve) been the main concern of silviculture since its foundation!

L299: finally, what is the monitoring period for the study? 10 years? 20 years? Else?

Fig1a: full red squares not specified. Further, only clear footprint of a saturation given by open red squares.

Fig4: only the first column is informative of a threshold. Perhaps needed to re-inform about the lack a sufficient LAI gradient on the right (but may LPJ-guess be pushed toward extrapolation anyway?)

L324: A bit puzzling that the 1st paragraph seeks to establish this result and ground it in the recent literature, with no reference at all to the classical forestry literature. Could this be a little bit better balanced, for the sake of enlarging the audience?

L335-339: how may this be evaluated properly? Are there any data possibilities available to support the issue?

L377-379: yet what allows to justify this statement? It may be conversely expected that the saturation point is reached at lower LAI for light-demanding tree species, in line with the clear forest management strategy. Not?

L399-401: and also, this has been observed in empirical studies of forest yield, whereby an optimum, instead of a plateau, has also been detected.

I am a bit puzzled with two conclusions:

*Above its saturation value of ~ 4 m2 m-2, additional increases in LAI are not linked to increased productivity, but may contribute to other functions selected in evolution, such as competition with adjacent trees, resource storage and buffering against herbivory*

Then, should this additional LAI have adaptive functions, some of these being adaptations to biotic or abiotic pressure of the environment, should we understand that – while decreasing LAI may be of minor impact on the strength of the C sink – it may impact stand/tree resilience to abiotic/biotic disturbances. Isn't it important as well? Shouldn't it be discussed?

*This threshold can be used to define sustainable metrics for sustainable harvesting, as 448 those that do not impact the carbon sink strength of the forest stand.*

And how? When this plateau was established in terms of ANPP = f (stand density), at least direct targets could be formulated for the prescription of silvicultural regimes. Yet, how may a forest manager easily pilot a LAI target?

Jean-Daniel Bontemps, Laboratory of forest inventory, Nancy, France, 23rd  January 2025

**Statement of conflicts of interest**

The author declares to maintain a recurrent professional relationship with the first author of this paper. Due to its interest in forest dynamics (demographic, not functional), he first volunteered a few remarks about the paper, which the first author considered worth of interest to be posted publicly in the discussion of this paper.

---

## Author Comment (AC1)

**Saturating response of photosynthesis to increasing leaf area index allows selective harvest of trees without affecting forest productivity**

Olivier Bouriaud, Ernst-Detlef Schulze, Konstantin Gregor, Issam Bourkhris, Peter Högberg, Roland Irslinger, Phillip Papastefanou, Julia Pongratz, Anja Rammig, Riccardo Valentini, and Christian Körner

**Authors' response to**
RC1: 'Comment on egusphere-2024-3092', Anonymous Referee #1

General comments:

The authors of this study investigated the impact of harvesting on the fluxes of carbon in forests over a large gradient. Based on eddy covariance measurements and on modelling approach, the authors explored the hypothesis that below a certain value of LAI, any forest management action such as harvesting or pruning does not affect Net Ecosystem Productivity. On the basis of a non-linear relationship between gross primary production (GPP) and leaf area index (LAI) characterised by saturation above a threshold of 4-5 m2 m-2, they concluded that above this value, the reduction in leaf area (due to forest management) therefore has little effect on net CO2 uptake and that it remains constant after partial harvesting.

Overall, the study is well structured and of great interest. However, I would suggest some major revisions, detailed below.

Major comments:

With regard to LAI values, it is difficult to understand whether the threshold value indicated by the authors is relevant whatever the PFT. In fact, the definition of LAI varies between deciduous and coniferous stands, due in particular to a difference in clumping index. As a result, its impact on carbon fluxes can also be expected to be different. This point deserves to be discussed. In addition, the results based on the analysis of carbon fluxes measured by eddy covariance technique should be further discussed in the light of the 'known' uncertainties concerning the estimation of GPP and Reco during the day.

> We thank you very much for reviewing our paper, his appraisal, and for providing these insightful comments or critics.

In general, it is difficult to assess the contribution of using the LPJ-GUESS model. This tool was mainly used to confirm the non-linear relationship between GPP and LAI and to confirm the LAI threshold value, but it could have been used to go further in analysing the weak impact of forest management (competition for light, for example).

> This is a very valid point. We used the model specifically to look at the LAI-GPP-relationship for different cohorts. This allows us to address one important shortcoming of the observations, namely that we do not know what GPP would have been if LAI were at a different value than observed. It also enables looking at the age cohorts at the same time step (which is what we did in Fig. 4), which allows us to exclude further factors like $CO_2$ fertilization that influence GPP and LAI over time.

> We do acknowledge that the model could be used to investigate other things, but as written in our answer below, the model was not designed to capture detailed site-specific aspects of observation sites. Furthermore, the way we set the simulations up was to understand the GPP of tree cohorts of various LAIs at the same site for the same climatic and atmospheric conditions. From the plots in Figure 4 we can see that additional LAI for an age cohort will not increase GPP after an LAI of around 4. However, the total LAI of the simulated area (which is the weighted average of all the cohorts standing there) did not exceed this threshold by much. In response to your comment, we will now add these total LAI and total GPP numbers to the plot as well and describe it better, since we indeed did not explain this very well. The model will actually prevent high stand LAI through self-thinning in temperate regions, which is why we resolved this by looking at the age cohorts.

> We agree that such an analysis regarding the relationships among simulated attributes and fluxes would be very interesting and important. However, this would require a substantial number of additional model simulations, which would likely be enough material for separate publication.

> We acknowledge this idea of the reviewer and will add a statement regarding next steps to the discussion, as well as further explanation on the choice of the modeling.

Specific comments:

151-153: For the sites studied, are the age and forest management of the plot described, and how have these characteristics been taken into account in the analysis?

> Indeed, the referee makes an important point, namely that several eddy sites neglected the auxiliary data. We had deleted sites, because of missing data.

> In as much as they are described we used the available data. the forest management in particular was described. The management type is described in Table S1 , and the harvest operations are being listed up and quantified in terms of biomass decrease in S 3.

162-166: For estimating LAI based on remote sensing, is the spatial resolution of MODIS images sufficient, particularly in relation to the size of the plots (consistent with the comparison with carbon flux measurements), to detect differences in LAI between managed and unmanaged sites?

> This is a very interesting point. The footprint of the EC measurements is a lot larger than the MODIS pixels (about 1 km$^2$: see Aubinet, 2012). This is coherent with the fact that the processes analysed in our study are inherently large-scale processes (as opposed to fine-scale, i.e., tree level). This is also the only meaningful scale to consider the effects of harvesting since partial harvesting is creating heterogeneities at fine scale but we only consider the overall stand-level consequences over the EC footprint.

> We were more concerned by saturation in LAI prediction from remote-sensed products. In our revision, we will pay attention that this is made clear.

223-227: In general, it is difficult to assess the contribution of using the LPJ-GUESS model in this study because the description is not very detailed: how is competition for light taken into account, in particular as a function of tree density, the age of the tree stand, etc.? how do photosynthesis parameters vary as a function of age, as a function of PFT? how does a reduction in soil water impact photosynthesis and/or production?

> We thank the reviewer for pointing out that we did not adequately explain the LPJ-GUESS processes. As the LPJ-GUESS model is only applied without modifications in this study, we wanted to avoid redundant explanation of its processes. However, we should have made clearer where such information can be found and more explicitly refer to the comprehensive description in Smith et al. (2014). We will consequently add more detailed descriptions in the supplements that are relevant to the simulations we conducted for this manuscript plus some sentences regarding the main processes in the main manuscript.

> Competition for light is based on weighting the leaf area (or LAI) of each simulated tree cohort, i.e. a tree cohort that has more leaves within a canopy layer will get more light. A cohort with a higher tree density can have a higher canopy leaf area leading to more light access. With rising age, the cohort mortality probability increases, reducing the tree density and thereby also the canopy leaf area. Photosynthesis parameters do not vary with tree age but are influenced by many other environmental and physiological parameters, for example, the maximum carboxylation rate $Vc_{max}$ is a function of leaf nitrogen content.

> A reduction in soil water content induces stomatal closure via a PFT-specific parameterization. Stomatal closure will then in turn lead to reduced levels of photosynthesis and subsequently lower growth.

228-230: Does this mean that carbon allocation is only calculated on an annual time step in the model? There are seasonal dynamics that affect the respiration rate associated with organ growth and therefore the NEE. This point needs to be clarified in relation to the conclusions of this study.

> Yes, allocation is only calculated at the end of the year. However, photosynthesis and respiration and therefore also NEE are calculated daily and accumulated towards the end of year. By doing so, seasonal dynamics are taken into account, for example NPP can become negative during the dry seasons of the year due to closed stomates but maintenance respiration still occurs. We acknowledge that not having seasonal or daily allocation is still a limitation of the model, but accumulating on an annual basis has been proven a reasonable simplification for many forests.

> Furthermore, we here only compare the model to yearly values of observed GPP, therefore we find this appropriate.

230-232: Does the SLA vary with position in the canopy (profile of SLA?) relative to leaf exposure to incoming radiation? This is an important point to take into account when considering light competition and its impact on NEP in relation to tree density.

> Unfortunately, the eddy flux measurement observes the canopy as a whole. Thus, the Flux sites do not cover this aspect. Experimental data may be available, but probably only in the form of experiments restricted both in time and space. However, we think that this would not change our analysis which refers to a "stand" treatment.

> From the modelling perspective, the model used belongs to the family of the big-leaf models. It hasn't been parametrized to reproduce within-canopy SLA variations. Moreover, SLA seems to be highly influenced by the light conditions themselves, so we could expect that a single parameter is not a viable option. Thus, we cannot include this in our analyses.

> In our anticipated revision we have to make this limitation clearer.

237-238: How does clumping index vary between PFTs, stand age and tree density? Is this variation taken into account when analysing the results?

> Thank you for bringing this point, which is will be discussed in the anticipated revision. There are two important elements regarding the clumping, which we anticipate to include in the manuscript: i) LAI estimations on the flux sites already take the clumping into account and ii) the spatial scale considered here is not easily connected to clumping.

The LAI on the monitoring sites was estimated based on hemispherical photographs. It also appears that the site managers have long studied the parameters necessary to do the estimations of LAI, and have estimated a clumping factor. So, while we fully agree that leaf clumping is an important factor when estimating LAI, but that it has been accounted for in our measurements.

About the second point, the canopy structure being a highly variable feature within a single stand, it is probably difficult to isolate this as a parameter or a driver. The difference between PFTs is visible in Figure 1 (subfigure showing **GPP = f(LAI)**).

We hope that our anticipated changes to the paper will satisfy the concerns by the referee.

267-268: Is the threshold of 4.5 m²/m² the same regardless of the clumping effect? Is this value the same for coniferous stands? Generally speaking, there is no discussion of the definition of LAI for a deciduous stand and that for a stand of conifers (see lines 285 & 305-307).

Yes, we believe that 4.5 $m^2m^{-2}$, while not being necessarily a unique and fixed threshold, likely represents an upper bound valid for all temperate forest types, including both, coniferous as well as broad-leaved forests. The main reason for this is that the non-linear relations among processes concerning the light absorption, the photosynthesis and the gas exchange were observed consistently in all forest types.

As shown in Fig. 1., the threshold is common to PFTs, one that integrates to a large extent the clumping effect. The data of coniferous and broad-leaved forest are well mixed. There is no difference detectable.

In the anticipated revision we will address this problem by highlighting the fact that the underlying processes were shown to be independent of the forest types.

273-275 & 394-396: This result is relatively expected because if the LAI value increases, we expect an increase in biomass (linked to an increase in canopy photosynthesis) which leads to an increase in growth respiration, one of the two components of autotrophic respiration. Why not use the model to deeply analyse the differences in partitioning of the two components of autotrophic respiration (respiration due to the energy cost of tissue maintenance and respiration due to the cost of tissue construction during the growth phase) between sites and forest management to confirm the hypotheses proposed by the authors? Can the model support the hypotheses mentioned, particularly with regard to the non-linear relationships found with GPP, the distribution of NEE between GPP and Reco, and even the distribution of Reco between growth respiration and maintenance respiration?

The photosynthesis model used in LPJ-GUESS is based on Collatz et al. 1991 which is a simplification of the Farquhar et al. 1980 model and the carbon allocation model based on

Smith et al. 2001. It would indeed be interesting to disentangle the types of respiration through modeling, but in LPJ-GUESS, growth and maintenance respiration follow are expressed via a simple relationship, therefore this is not possible.

```
Growth respiration = 0.25 (GPP - maintenance respiration)
```

Regarding the other aspects, we will add further insights on the modeled vs observed fluxes, see also our answer below.

417-418: Yes, a discussion on the uncertainty of the GPP estimate could be added, as well as for Reco values during the day (see also lines 304-305). The impact of the age of the stands selected for this study on the growth respiration rate in terms of the amount of living tissue (not total above-ground biomass) should be discussed. An increase in growth respiration could also be expected if there is a stand management practice such as pruning.

We acknowledge that these are interesting points that would allow to go deeper into the processes and feedbacks that stand beyond the responses that we observe in the experimental data. The partitioning into Ra and Rh bears additional uncertainties, which we cannot resolve with the existing data. This is future research, maybe not possible on all sites, depending the on the availability of parameters.

420-421: Why didn't the authors try to validate the model's predictions of NEE, GPP and Reco on these two sites? Once this had been done, the model could have been used to validate the hypothesis of an equilibrium LAI and to confirm the threshold value of 4.5 m²/m², and to test the impact of a change in the clumping index due to forest management.

This is a reasonable point. However, the problem is that LPJ-GUESS is not designed to perfectly capture highly site-specific properties. We aligned the simulations in this study with the observations to get model results also for the various climates, soils, and species types. Capturing the exact details of a site, including exact age distributions, and management impacts, would require detailed data for the sites that are not available and even then probably not capture the exact properties of the sites. We used the model here to show the non-linear response of GPP to LAI.

In fact, the default management scheme in LPJ-GUESS is based on executing thinning when LAI gets above a threshold, therefore this cannot be used for this experiment.

Nevertheless, we will add further validation of the model (see also our answer below) and try again whether we can include more model results to back the claim for management as well.

Fig 1: The GPP/LAI relationship is difficult to interpret due to the high variability of GPP values (e.g. for managed conifer/mixed). No point corresponds to the case of managed broadleaves (mentioned in the legend). For the Reco/LAI relationship, it would be interesting to indicate the uncertainties on the graph in the same way as for the GPP/LAI relationship.

> The managed broaleaved sites are present in the figure (filled diamonds). There are 5 such sites in our analyses. We will improve the readability of the figures for the revision.

> The imbalance in the sites (managed/unmanaged, conifer/broadleaved) is an important current issue. The modelling intends to bridge the gap. It largely confirms that the relationship observed would hold in many other situations. Uncertainties are indeed quite large in fluxes.

> Also, proper management operates near the self-thinning line (Luyssaert et al., 2011, see above). Therefore, the mixture of sites strengthens our argument, that a fraction of growth can be harvested without significantly impacting the fluxes.

> In the anticipated revision we will carefully check that this point is made clear.

Fig 3: as for figure 1, it would be interesting to identify coniferous sites from broadleaves sites.

> Certainly, this is a good point and one that is totally feasible.

Fig 4: Why not show the measured NEE in addition to the simulated NEE?

> Thank you for this suggestion! The reason was that we plotted the LAI and Fluxes of all age cohorts that were simulated by the model, leading to one dot per age cohort. For this, no measurement data is available, only for the entire site. Our plot shows the LAI-GPP relationship of cohorts of various ages and thus LAI, while all other factors (e.g., climate and co2) are equal, since they are simulated at the same site for the same year (we will make this more specific in the figure caption).

> However, we can compare the total fluxes with the modeled total fluxes and add these to the plots. Below is a simple example showing two additional plotted points for observed and modeled total LAI and GPP (we will then come up with a more detailed plot regarding uncertainties and so on, this is just as an indication).

---

## Author Comment (AC3)

**Saturating response of photosynthesis to increasing leaf area index allows selective harvest of trees without affecting forest productivity**

Olivier Bouriaud, Ernst-Detlef Schulze, Konstantin Gregor, Issam Bourkhris, Peter Högberg, Roland Irslinger, Phillip Papastefanou, Julia Pongratz, Anja Rammig, Riccardo Valentini, and Christian Körner

**Authors' response to a comment on: Saturating response of photosynthesis to increasing leaf area index allows selective harvest of trees without affecting forest productivity**

This contribution is an important one, in that it provides a functional insight into a long- attested empirical demographic relationship relating plant/tree population density and yield. In spite of its robustness, this fundament of yield science and silviculture (being able to extract trees from a forest without sacrificing total yield, and increase growth of remaining trees in a process that fastens the acquisition of size, thus preserving its long-term sustainability)
remains poorly perceived by ecologists and the general public, causing ill-posed debates in forest sustainability. For this reason, being able to support it by functional process-based grounding is major. This is an originality of the contribution.

> Thanks for your general conclusion. This is exactly why we wrote this paper, and we are happy to make any changes that makes this paper stronger. Indeed, the public has bin indoctrinated that harvest is bad. We did not speak this out, but our paper intends to give a functional basis for harvest that has been overlooked in the past: the role of leaf area.

That said, we also wish to share some matter of concern regarding specific aspects of the contribution, in order to give it the broadened perspective it deserves.

1. Constant final yield

One first aspect is the apparent disconnection there is, between this "modern" approach relating C fluxes and NEP from the EC methodology, and the bunch of historical work performed in forestry science to relate yield (ANPP) to stand density, also in agronomy science. The same saturation has been described as "law of final constant yield", "Langsaeter's plateau", the "thinning response hypothesis". Major texts include Yoda et al 1963, Assmann 1970, Pretzsch 2009, but also and more recently the nice synthesis by Weiner and Freckleton 2010 (named "constant final yield").

> ☆ Reference to these texts and concepts would allow better connect these findings to a long-established reality.

> We thank you for this suggestion. Our manuscript already contains many references, more than 70. But we indeed could make more reference to existing forestry literature. You are totally right, that we are talking about the same "rules" of management, and confirm "old" knowledge with process-based parameters.

> If we are allowed to submit a revision, we will extend the text accordingly.

This also raises an issue. The following conclusive statement: This threshold can be used to define sustainable metrics for sustainable harvesting, as those that do not impact the carbon sink strength of the forest stand questions how it can be achieved. When this plateau was established in terms of ANPP = f (stand density), at least direct targets could be formulated

for the prescription of silvicultural regimes. Yet, how may a forest manager easily pilot a LAI target?

⭐ Here, I wonder whether plot density data are available, and to what extent the correlation between LAI and stand density (say BA, or N) would allow capture/confirm the alleged LAI threshold invariant identified, and may pave the way toward these metrics. I would rather incline toward acknowledging the merit of this transposition of initial stand density / yield relationship to LAI / photosynthetic C fixation to ground the previous relationships, and deliver the message that forestry science is justified to do so. Perhaps the aspect may be expanded both in the introduction and discussion.

Thanks for this suggestion. To our knowledge the stand density data are not available as auxiliary data at the moment. Further, the relationship between LAI and stem density is weak or even inexistant:

https://infodoc.agroparistech.fr/index.php?lvl=notice_display&id=95892

For the time being, we only could make the point that future research is needed to translate LAI into a forestry scale. Even though it is not quantitative, forest managers know, if a stand is getting too dense. In this paper it is our aim, to give a functional reason to this empirical knowledge.

2. Tree species diversity and species traits

As written, the paper gives an impression that finding the threshold is so motivating that the issue of variability is a bit discarded. At L277-279, it is even suggested that cross-species variation may be minor, but no reference is suggested for this aspect. By contrast, I would be of the primary opinion that the shade-tolerance of tree species may influence that potential threshold, light-demanding species meeting an earlier LAI threshold than others. This would be consistent with the notion of clear-forest silviculture as envisioned in the 19[th] century for the management of light-demanding species (e. g. pines, oaks).

⭐ May it be useful to discuss the invariance of this LAI threshold in view of species traits? How far can we go? What may be the implications in terms of experimental designs based on current tower flux facilities? Is it an issue for a near future, or currently out of reach? This discussion would be welcome in the discussion.

We suggest that we extend the text and cite Luyssaert et al (2011) who showed that proper management operates very close to the self-thinning line, independent of tree species. The self-thinning line operates along different stand densities. Thus, this would be a way to translate our "theoretical" paper into forestry operations.

The variability of the threshold, for instance depending on the dominant species and biophysical parameters is largely unknown. We could not go any further with the current data. The uncertainty around the values of that threshold, and its variability, need to be addressed before it can be used as a management tool.

Further, may the design structure of EC sites allow to explore, at least a bit, the issue, e. g. by ranging sites according to some community weighted mean of the shade-tolerance trait (TRY) and see whether it correlates to some parameter related to the fit of the non-linear responses to LAI? I have this impression that a prescriptive unique threshold value should be avoided.

✪ Is it possible, or prohibitive, to explore a quick statistical relationship between the position of the threshold and a community weighted-mean of shade tolerance?

Thank you for this comment. We tried to use the TRY-database, but did not come to a useful result. The problem is, that, with stand growth, we operate at different stand densities. The "beauty" of using eddy-fluxes is, that it integrates over the whole canopy.

We suggest, that if we are allowed to submit a revision, we will make this point clearer.

3. The role of excess LAI

In the discussion, the adaptive significance of "excess LAI" is discussed, which suggests that excess LAI may serve the purpose of better tree resilience to abiotic or biotic disturbances. If so, while saturation is good news for the C storage, it is not necessarily the same for other functions related to resilience, in the present environment. Since it is indicated that some monitoring plots experience heat/drought or other disturbances over the period, may this be tested, or is it out of reach?

✪ I am uncertain about whether this test may be easy. At least, a better balance of the corresponding paragraph, and the trade-off it may induce in a silvicultural approach, is welcome.

Thank for this comment. We only discuss the role of excess LAI. To our experience it has a role only in competition. It has a cost of putting these leaves into position and to maintain them with water and nutrients. Thus, we do not think that they contribute to resilience. Management basically replaces natural competition. Unwanted competitors are taken out, favoring the target species/individual.

4. Methodological aspects

In general, it is quite hard to grasp the monitoring period under study, precisely. And also to compare it to management events, in order to perhaps question the ranking of managed/unmanaged plots. It may seem worrisome that only 11/19 plots classified as "managed" did not show any management event over the monitoring period in view of the rapidity of LAI reconstitution in general.

✪ May this have contributed to dampen the difference between managed / non-managed forests? Shouldn't this be discussed a bit (and if possible, tested? And if untested, what would be the perspectives ?)

To our feeling there is some bias, because it is cumbersome to harvest next to the tower. Timo Vesala, Finland, can tell stories about hand-carried timber. Thus, in managed forests there is the tendency by managers not to come next to this monitoring tower.

Thus, we are not surprised that the majority of tower sites remained untouched, even though the forest remains a managed structure. Maybe flux net has to move towers periodically, as we suggested in our paper published in Annals of Forest Science (Schulze et al, 2022: https://infodoc.agroparistech.fr/index.php?lvl=notice_display&id=95892). If we are allowed to submit a revision, we will clarify this point.

It was also very difficult to me to understand whether GPP/NEP data come from field plot measurement, or from LPJ-guess, and also what specific role does the model indeed play.

✪ I have this impression that the modelling objective may be specified more clearly in

the dedicated section, and early in the introduction. E. g., is it because it is feared that the integration scale difference between plot and EC towers may have a role?

All our data come from field measurements. We used the model only to show that the field observations can be reproduced by modelling and to compensate the imbalance in managed/unmanaged or forest types (conifer/broadleaved/mixed). This is important to show that we did understand the main processes.

Your concern is well taken. We must make clear that the datapoints are measured data.

**5. Local detailed comments**

Line 78-79: Bontemps 2021 (Plos) does not suggest any saturation of the carbon stock in European forests.

At stand level there exists a limit, the increase in stock is not indefinite. Trees must make a new tree-ring in order to survive. But the risk of failure increases with volume and canopy height. This is why even protected stands eventually level off (Nagel et al 2023). If we are allowed to submit a revision, we will clarify this point.

Line 95-98: the saturation of forest productivity with stand density has been a cornerstone of silviculture since the elaboration of the scientific principles of forest management. It is a bit frustrating it is not mentioned a bit more in-depth, a law known as the thinning response hypothesis, Langsaetter's law, and more generally in agricultural yield science, as the law of final constant yield.

We agree that our paper should make a better link to earlier observations, where the law of final constant yield is a key observation, and our observations and modelling results are very well in line with this law. If we are allowed to submit a revision, we will clarify this point.

In particular, Saturating response of photosynthesis to increasing leaf area index allows selective harvest of trees without affecting forest productivity may be rephrased as Saturating response of forest ANPP to increasing stand density allows selective harvest of trees without affecting forest productivity.

Line 122: why it would have impact on $CO_2$ assimilation is less clear.

We cited Monsi and Saeki. We will try to edit this part to make it more clear

Line 128: and certainly also, across tree species, given that their sensibility to light (shade-tolerance) may have a contrasted effect on a unique LAI threshold.

We guess, there is a limitation to deal with shade tolerance in our paper, because the eddy-flux measurement derives a bulk flux of the whole canopy.

We will refer to Luyssaert et al. (2011) where the main observation was that proper management is close to the self-thinning line

Line 133-135: In particular, the interactions between management and LAI, and their consequences for the carbon sink strength need to be determined in order to examine the consequences of wood harvesting on forests carbon sink strength.

> This is exactly the focus of our paper, and we believe that the importance of LAI in the relation between management and carbon sink strength needed to be brought back into discussions.

+ line 138-140: explore the possibilities of defining levels of sustainable partial cuttings from the perspective of carbon 140 fluxes, key to designing forest managements strategies

But how in practice, can we transfer a target N trees / ha under management into a LAI reduction / ha, especially since selective thinning can be deployed for different strategies (thinning from above / below)?

> Thinning from above and from below have the same target, namely to regulate biomass/ha, i.e. stem density. Which thinning method gets used is related to the anticipated wood use. In spruce, thinning from above provides a large stem volume based on the assumption that thin stems can resume growth.

> In beech, the early thinning is also from above, taking away stems with multiple branches (low quality wood) and later we support the dominant stem for higher growth.

> In both cases Luyssaert et al (2011) showed that proper management operates close to self-thinning. Here we show that the distance to self-thinning may be the extra leaf area that can be taken out without harm.

> Nevertheless, as stated at an earlier comment, there may not be any relation between stem density and LAI.

Line 154-155: what is the monitoring period exactly? Is it 2000-2020 as suggested by LAI measurements? If so, one may wonder about the significance of being classified as managed forests for those 8 study sites out of 19 that were matter of no harvesting over the period.

> In our Annals-Paper (Schulze et al., 2022 op. cit.) we show how management "jumps" across the property. Thus, it is possible, especially in older stands, that no harvest takes place in a given 20 yr period, e.g. in oak, and if there is a measuring tower (see above).

Line 164-166: were these MODIS-based estimates compared against measurements for those sites where site measurements for LAI do exist?

> Yes, the values are presented in the supplementary table 1. As a regrettable limitation to this comparison, the LAI values do not correspond to the same year or period in time. This is one of the reasons why we advocate for more consistent and frequent measurements throughout the network.

Line 168-173: in addition, may it be possible to specify a bit across which spatial range do the flux measurements integrate and to what extent there are representative of fluxes within the perimeter of the study sites?

> The actual footprint of eddy covariance towers remains difficult to determine, but are in the order of 1 km² (see Aubinet, 2012). We are not aware of any publication that would present in a standard way the current estimation of the footprints.

Line 183-185: then if GPP and Reco are gaussian, why the difference = NEE would not be so? Are their discrepancies to be expected in this budget?

> GPP and Reco are not calculated from eddy covariance recordings the same way. They both involve several steps which are only partially independent. Both compensation and amplification may occur.

L 203: but also suppressed trees, with varying LAI

> Your point is well taken, but we are afraid we cannot deal with all aspects of the "real world" of forest management. For example, in broadleaved stands we leave suppressed trees, because they do not harm, and shade the trunk of the dominant target tree, and it would cost work-time to cut them down without any use.

> Obviously, a technical paper is needed to relate the findings of this study to forest management

Line 209-210: then how can it be that measured fluxes are representative of forest plot dynamic?

> Indeed, the flux network replaces time by space. In the study of Luyssaert et al (2021) about old-growth forests it was made visible how fluxes explain stand dynamics. We will refer to this study.

Line 218: on monitoring plots, wasn't it possible to more directly measure tree-driven GPP, or at least compare?

> Some stations measure GPP with chambers as well, but these data were not continuous and not available for all sites. The amount of work involved is probably discouraging, given that multiple trees would need to be monitored in order to obtain reliable per-area figures.

Line 145-246: at this point, the issue of variability across tree species turns an issue. What are the species covered in the 30 monitoring plots? How do they match the PFT of LPJ-guess? To what extent there is a risk to remove this variability?

> The list of the dominant species can be found in the site description, and is described in the supplementary table. It is probably not so much the variability of species that creates issues in detecting management effects on carbon fluxes, than the fact that the replication is very low, and that the current data are not balanced in regard to the PFTs. We acknowledge this in the manuscript and advocate for a more balanced sampling of the PFTs and a better representation of management.

L253-254: can it be specified whether the management operations described on the monitoring plots were introduced into LPJ-guess? Or else?

> Management operations did not occur on all plots. The objective of the modelling was not to reproduce these operations, which would require a lot of parametrization work. This could be the object of a further study.

L262-263: I would say this (should) has(ve) been the main concern of silviculture since its foundation!

> Indeed, but on a volume/ha basis, and not explained by fluxes. We will make this link if we are allowed to submit a revision.

L299: finally, what is the monitoring period for the study? 10 years? 20 years? Else?

> Please refer to the Supp table 1.

Fig1a: full red squares not specified. Further, only clear footprint of a saturation given by open red squares.

> Sorry that our figure caption was not complete. In a revised paper we will take care of this.

L324: A bit puzzling that the 1st paragraph seeks to establish this result and ground it in the recent literature, with no reference at all to the classical forestry literature. Could this be a little bit better balanced, for the sake of enlarging the audience?

> We like to thank for this comment. See above, we will make this link in a revision

L335-339: how may this be evaluated properly? Are there any data possibilities available to support the issue?

> Yes, it could happen in future, if harvest was recognized by the micro-met people. We addressed this in Forest Ecosystems (Schulze et al., 2021)

L377-379: yet what allows to justify this statement? It may be conversely expected that the saturation point is reached at lower LAI for light-demanding tree species, in line with the clear forest management strategy. Not?

> We agree, the link to species traits and overall behavior needs to be established. At the moment we believe that the data available is insufficient for such objective. We are left to hypotheses, but this one seems quite reasonable.

L399-401: and also, this has been observed in empirical studies of forest yield, whereby an optimum, instead of a plateau, has also been detected.

> The empirical studies of yield suggest an optimum in relation to the stem density or the standing biomass. It is unclear if this would necessarily translate into an optimum in relation to LAI because LAI is not related to either stem density or standing biomass. It could be hypothesized however that, in some particular situations, a response to LAI would display an optimum instead of a plateau.

I am a bit puzzled with two conclusions:

Above its saturation value of ~ 4 m2 m-2, additional increases in LAI are not linked to increased productivity, but may contribute to other functions selected in evolution, such as competition with adjacent trees, resource storage and buffering against herbivory

Then, should this additional LAI have adaptive functions, some of these being adaptations to biotic or abiotic pressure of the environment, should we understand that – while decreasing LAI may be of minor impact on the strength of the C sink – it may impact stand/tree resilience to abiotic/biotic disturbances. Isn't it important as well? Shouldn't it be discussed?

We suggested that a fraction of the leaf area is not contributing to increasing the productivity and that it may serve other functions. We agree that the resistance to pathogens (i.e., having spare leaves) could be one such function.

This threshold can be used to define sustainable metrics for sustainable harvesting, as those that do not impact the carbon sink strength of the forest stand. And how? When this plateau was established in terms of ANPP = f (stand density), at least direct targets could be formulated for the prescription of silvicultural regimes. Yet, how may a forest manager easily pilot a LAI target? 448

Probably, this is future research. These days a forest manager can get satellite LAI from public data, but this was never been suggested that foresters observe LAI.

In a more technical oriented paper, we could address all the suggested concerns of "real" forest management

Jean-Daniel Bontemps, Laboratory of forest inventory, Nancy, France, 23$^{rd}$ January 2025

Statement of conflicts of interest

The author declares to maintain a recurrent professional relationship with the first author of this paper. Due to its interest in forest dynamics (demographic, not functional), he first volunteered a few remarks about the paper, which the first author considered worth of interest to be posted publicly in the discussion of this paper.

Scientific discussions are always welcome and often lead to valuable insights. We truly appreciated the constructive exchange of ideas, which helped refine certain aspects of this work by highlighting key findings as well as remaining questions and caveats. We hope these discussions and debates will be useful for future research.

---

## Author Response (AR1)

**Saturating response of photosynthesis to increasing leaf area index allows selective harvest of trees without affecting forest productivity**

Olivier Bouriaud, Ernst-Detlef Schulze, Konstantin Gregor, Issam Bourkhris, Peter Högberg, Roland Irslinger, Phillip Papastefanou, Julia Pongratz, Anja Rammig, Riccardo Valentini, and Christian Körner

**Authors' response to**
**RC1**: 'Comment on egusphere-2024-3092', Anonymous Referee #1
and
**RC2**: 'Comment on egusphere-2024-3092', Anonymous Referee #1

Again we are very thankful for the reviewer's suggestions and for the editorial work.
The major changes underwent were:
- shortening the introduction according to RC2's suggestions and critics, which is now more focused;
- made the necessary complements to the methods regarding the LAI estimation, to the fluxes and the model;
- redrew the figures 1, 3 and 4;
- made several changes to the discussion in order to incorporate the elements suggested, such as the uncertainty in LAI and fluxes, uncertainty in the location of the LAI threshold and its variations among plant functional types (or other factors);
- discussed the limitations of the model;

**Modifications made in response to the Major comments by RC1:**

With regard to LAI values, it is difficult to understand whether the threshold value indicated by the authors is relevant whatever the PFT. In fact, the definition of LAI varies between deciduous and coniferous stands, due in particular to a difference in clumping index. As a result, its impact on carbon fluxes can also be expected to be different. This point deserves to be discussed. In addition, the results based on the analysis of carbon fluxes measured by eddy covariance technique should be further discussed in the light of the 'known' uncertainties concerning the estimation of GPP and Reco during the day.

> We have highlighted the existence of variations among PFTs in the threshold in the results section. We also highlighted the uncertainties in both the LAI and the fluxes estimates, present them in the Figure 1, in the text and discussed these uncertainties in the discussion section.

*In short, we suggest that the location of the threshold is not precisely determined by our data, and thus present a range rather than a single value, and mention the possibility that this threshold varied among PFTs as suggested by the data and the model.*

In general, it is difficult to assess the contribution of using the LPJ-GUESS model. This tool was mainly used to confirm the non-linear relationship between GPP and LAI and to confirm the LAI threshold value, but it could have been used to go further in analysing the weak impact of forest management (competition for light, for example).

*The model was used in order to verify that it could represent a saturation in the responses of GPP and NEP to increasing LAI. It confirmed this and also highlighted the variation among sites in the location of the threshold value. We inserted some lines to better explain the expected outcome of the current modelling work, and, in accordance to your very pertinent suggestion, added the total LAI and total GPP numbers to the plot (with their uncertainties).*

*We added a statement regarding next steps to the discussion, as well as further explanation on the choice of the modeling (L507-515).*

*We lately found a satisfying way to model management, but could not incorporate it in this revision because we lacked the time to do so. It allows to compare managed/unmanaged time series of LAI, GPP and NEP, and could be added to the manuscript at a later stage.*

Specific comments:

151-153: For the sites studied, are the age and forest management of the plot described, and how have these characteristics been taken into account in the analysis?

*We made no particular changes.*

162-166: For estimating LAI based on remote sensing, is the spatial resolution of MODIS images sufficient, particularly in relation to the size of the plots (consistent with the comparison with carbon flux measurements), to detect differences in LAI between managed and unmanaged sites?

*We tried to make it clearer in the text, that the spatial scale of the study is that of stand, the typical footprint area of eddy covariance being in order of 100 ha (L272-274).*

*We highlighted the uncertainties concerning the estimation of LAI both in the text and the figures.*

223-227: In general, it is difficult to assess the contribution of using the LPJ-GUESS model in this study because the description is not very detailed: how is competition for light taken into

account, in particular as a function of tree density, the age of the tree stand, etc.? how do photosynthesis parameters vary as a function of age, as a function of PFT? how does a reduction in soil water impact photosynthesis and/or production?

> We provided more details to explain the LPJ-GUESS processes, for instance concerning its representation of the canopy, its internal photosynthesis model (L272-274, 282-286).

228-230: Does this mean that carbon allocation is only calculated on an annual time step in the model? There are seasonal dynamics that affect the respiration rate associated with organ growth and therefore the NEE. This point needs to be clarified in relation to the conclusions of this study.

> We introduced new sentences to disclose the functioning of the model (L282-288) and as suggested as discuss these limitations in the discussion (L510-515).

230-232: Does the SLA vary with position in the canopy (profile of SLA?) relative to leaf exposure to incoming radiation? This is an important point to take into account when considering light competition and its impact on NEP in relation to tree density.

> Variations in SLW could not be included in our current study. We explicitly mention this limitation L282-283.

237-238: How does clumping index vary between PFTs, stand age and tree density? Is this variation taken into account when analysing the results?

> We introduced several sentences to remind this problem of clumping. We present it in the Material and Methods section along with the reference by Gielen et al. (2018) where the estimation of the clumping index is described (L197-198). We discuss these problems further in the discussion, around the uncertainties of estimating LAI.

> Figures 1 and 4 now present the uncertainties in LAI and fluxes.

267-268: Is the threshold of 4.5 $m^2/m^2$ the same regardless of the clumping effect? Is this value the same for coniferous stands? Generally speaking, there is no discussion of the definition of LAI for a deciduous stand and that for a stand of conifers (see lines 285 & 305-307).

> We modified Fig. 1. in order to highlight the differences between conifers and broadleaved. We also discuss the matter, for instance L 431-434.

273-275 & 394-396: This result is relatively expected because if the LAI value increases, we expect an increase in biomass (linked to an increase in canopy photosynthesis) which leads to an increase in growth respiration, one of the two components of autotrophic respiration. Why

not use the model to deeply analyse the differences in partitioning of the two components of autotrophic respiration (respiration due to the energy cost of tissue maintenance and respiration due to the cost of tissue construction during the growth phase) between sites and forest management to confirm the hypotheses proposed by the authors? Can the model support the hypotheses mentioned, particularly with regard to the non-linear relationships found with GPP, the distribution of NEE between GPP and Reco, and even the distribution of Reco between growth respiration and maintenance respiration?

> The photosynthesis model is presented in the text L287-288. Further, we introduced new sentences to discuss the limitations inherent to the model (it being a big-leaf model, with no SLA variations throughout the canopy, and the lack of daily allocations).

417-418: Yes, a discussion on the uncertainty of the GPP estimate could be added, as well as for Reco values during the day (see also lines 304-305). The impact of the age of the stands selected for this study on the growth respiration rate in terms of the amount of living tissue (not total above-ground biomass) should be discussed. An increase in growth respiration could also be expected if there is a stand management practice such as pruning.

> We present uncertainties in GPP estimates and show them in the Figures 1 and 4. More on the effects of management could be done in the future, by comparing the simulated fluxes under management/no management.

420-421: Why didn't the authors try to validate the model's predictions of NEE, GPP and Reco on these two sites? Once this had been done, the model could have been used to validate the hypothesis of an equilibrium LAI and to confirm the threshold value of 4.5 m²/m², and to test the impact of a change in the clumping index due to forest management.

> This is a reasonable point. However, the problem is that LPJ-GUESS is not designed to perfectly capture highly site-specific properties. We aligned the simulations in this study with the observations to get model results also for the various climates, soils, and species types. Capturing the exact details of a site, including exact age distributions, and management impacts, would require detailed data for the sites that are not available and even then probably not capture the exact properties of the sites. We used the model here to show the non-linear response of GPP to LAI.
>
> In fact, the default management scheme in LPJ-GUESS is based on executing thinning when LAI gets above a threshold, therefore this cannot be used for this experiment.
>
> Nevertheless, we will add further validation of the model (see also our answer below) and try again whether we can include more model results to back the claim for management as well.

Fig 1: The GPP/LAI relationship is difficult to interpret due to the high variability of GPP values (e.g. for managed conifer/mixed). No point corresponds to the case of managed broadleaves (mentioned in the legend). For the Reco/LAI relationship, it would be interesting to indicate the uncertainties on the graph in the same way as for the GPP/LAI relationship.

We have redrawn the figure.

Fig 3: as for figure 1, it would be interesting to identify coniferous sites from broadleaves sites.

The figure 3 was redrawn and now shows the conifer/broadleaves sites.

Fig 4: Why not show the measured NEE in addition to the simulated NEE?

We added measured and modelled NEP in the figure 4 in accordance to this suggestion.

**Modifications made in response to the Major comments by RC2:**

54: value should have associated uncertainty for management guidance. Is 4 m2/m2 a minimum?

We made several changes that go in that direction, by indicating a likely range of values for the threshold, which could vary according to functional plant types and other factors. We also generally present in more details the uncertainties in the LAI and fluxes values. We have redrawn Fig 1 and 4 in this sense too.

57: note that this applies to temperate forests

We changed the text accordingly (L60).

67: 'counteracts climate change mitigation' sounds like a bit of a double negative and was confusing to read to start of the manuscript. I get it, but had to pause. The next sentence discusses mitigation rather than counteracting mitigation so one's mind is pulled in two directions.

Changed as planned.

71: who assumes this? I wasn't aware that this was a common perception amongst scientists, at least forest scientists.

 These parts were removed.

77: this isn't always the case e.g. https://www.nature.com/articles/nature12914 and 'very low' is at a minimum qualitative. There's a huge literature on this topic (https://www.nature.com/articles/nature07276) with lots of controversy (https://www.nature.com/articles/s41586-021-03266-z) as the authors are well aware and the statement as written discounts this rick literature. I've come to the opinion that people with forest management training think that old stands stop growing because monoculture forests basically do, but natural forests can keep taking up carbon even if at a slightly slower rate. Having been in temperate forests where the mid story is comprised of trees that we would think of as fully grown mature adults with overstory trees proper old growth giants, I've always been mystified at the idea that old growth forests don't still take up carbon. Obviously there is some physical limit. I don't disagree that older forests might take up less carbon, rather the assumption that they always do; for example take a look at the data points in Fig. 1 here instead of the curves that were hacked through them: https://assets-eu.researchsquare.com/files/rs-5183310/v1_covered_49c18487-5655-429a-a89d-b2e4d64fa22e.pdf?c=1735007962

 These parts were removed.

88: I somewhat that wood provision is considered a disturbance if sustainably harvested to simulate natural forest processes. The passage could easily be re-written to emphasize what the paper is actually about: that harvesting can occur with minimal disruption to carbon sink strength.

 We shortened and tried to focus as suggested.

95: disagree that selective harvesting is a disturbance or at a minimum that a disturbance is a bad thing; forest harvesting can simulate 'natural' forest processes as noted above.

 Revised accordingly (L116).

As a whole, the Introduction makes a number of valid points, but is weakened by assumptions and poorly-cited statements. It should be re-written to focus more strongly on the matter at hand, and can be guided around the LAI of 3.5 m2/m2 found by Schultze to expand this argument to carbon gain in addition to conductance.

 The introduction was shortened and is hopefully better now.

151: couldn't most of the Canadian sites from BOREAS be considered unmanaged conifers?

As mentioned we focused exclusively on temperate sites.

153: how was LAI estimated? I see some text on line 162 but this could be written in a much more systematic way for a methods section. Remote sensing and ground-based estimates should be compared to understand their differences (and both have substantial uncertainty that should be estimated if possible).

We introduced several sentences to remind this problem of clumping. We present it in the Material and Methods section along with the reference by Gielen et al. (2018) where the estimation of the clumping index is described (L197-198). We discuss these problems further in the discussion, around the uncertainties of estimating LAI.

168: the flux is transport across an area, so yes by measuring transport across the sonic anemometer and gas analyzer the eddy covariance system is physically measuring a flux. Calculating a surface-atmosphere flux does require some assumptions, I agree. Most instruments really just measure voltage differences so one could also argue that nothing measures a flux.

These sentences we rephrased (L203-205).

172: low ustar needn't represent an error in measurement, it just seeks to represent a case of insufficient turbulence where the assumptions that underlie the eddy covariance technique are not good assumptions. For all we know the sensor measurements themselves can be of the highest quality.

These sentences we rephrased (L204-208).

Figure 1: surprised that red and green are being used at the same time. Please use different choices for our colorblind colleagues.

The figures were redrawn correspondingly.

194: please cite the R package

Done, L242.

210: this can vary widely based on tower height and environmental factors; Chu et al. (2019) have the most systematic study of footprints across multiple sites and citing this study here can help clarify quantitative aspects of flux footprint dimensions at the network scale.

We do not present specific estimations of the footprint area, but mention its order of magnitude L275-277. It is unclear if each site has the same methodology to estimate the footprint, and if the area of the footprint could be a covariate in this study.

221: 'demonstrated' instead of 'proven' is probably a better verb here.

Done, thank you.

263: in the results section define what is meant by 'near'. The manuscript should be strengthened by including uncertainty estimates in multiple locations including here (also line 266, etc.).

Done and changes made to Fig.1.

267: yes, because it's not a location, it's a mean value with uncertainty. That could and should be quantified, either as a single value or a threshold that varies as a function of climate or forest type.

We revised Fig. 1 according to this suggestion, and show the threshold with its uncertainty. We also introduced new sentences to discuss the

279: if it's not significant, it didn't tend to be higher. But it could be higher in future studies with more statistical power perhaps.

Indeed, if it had been significant, it would have been significantly higher. Since it's not, only a tendency can be evoked. The discussion will highlight the fact that significance is not reached and we will call for more experimental data to increase the statistical power.

292: there are more significant digits reported than warranted for a study of dry matter at a plot scale.

The mistake was to use a dot instead of a space or a coma: the values are very high and not presented with decimals, expressed directly in g.

321: this is a great rule of thumb but adding an improved statistical analysis could further improve it, or note that the analysis points toward a rule of thumb that could be valuable guidance for foresters with additional research.

We introduced new sentences in the discussion in this sense L401-404 and 438-440.

331-332: add scientific names. Note also that where the authors are writing from there is a single type of Fagus, but another in the eastern Mediterranean (although the eastern European

one is now recognized as a subspecies), and quite a few species in Asia such that simply stating 'Fagus' might cause unnecessary confusion for an international study. Note also that Fagus is italicized on line 420 but not in other places.

> We revised the text accordingly.

Fig. 1: are these data points from the eddy covariance data or the model? If the former, what partitioning method was used to infer GPP and Reco?

> We made complements to the presentation of the flux methods (L).

Figure 3: is <10 etc. the age of the forest since last stand replacement or the time since last management prescription?

> The figure was changed to present distinctly the conifers and broadleaved. This new version is also less confusing than the duration presented in the earlier version of the figure.

Fig. 4: slightly larger font sizes would make this easier to read.

> Changed accordingly.

735: species name

> Changed accordingly.

---

## Author Response (AR2)

Dear Editor,

We are very thankful for your consideration and for the reviewer's suggestions. We made the changes requested by the reviewer, as we agreed with them (with one exception). The changes made are therefore:

- L39: there are more than 212 sites.
Indeed, even before 2015, there were more than 212 eddy covariance sites globally, but not all of them were included in the FLUXNET2015 release. Many of these sites were part of different network, contributed data separately, or did not meet the specific data requirements for FLUXNET2015. We modified the text as:
 In its 2015 release, FLUXNET represented 212 sites worldwide of eddy covariance.

- L140: what is long-term?
We inserted a text to explicit this: "i.e., > 10 years whenever possible" and thus suggest 10 years as being the threshold for long-term.
10 years seemed a reasonable threshold, given that eddy covariance is still a young method. In our selection set of sites, 19 over the 29 sites have an observation period > 10 yrs. The supplementary table ST1 discloses the exact series length.

- L182: Lasslop et al. 2008 is a study that one can use to argue that eddy covariance data uncertainty can approximate a normal distribution.
We included this citation in the manuscript and thus the reference too.

- L206: The EC footprint is less than 1 km$^2$
There are indeed some debates regarding the size of the footprint. To us it was only important to underline that the study is not a tree-level study (ie. ~ 100 m$^2$).
After some debates we decided to change the manuscript and mention a 0.1 km$^2$ footprint size.

- Figure 3 and elsewhere: note that the biospheric rather than atmospheric convention is being used for the sign of NEE (I may have missed it in the text).
Here we inserted a sentence to clarify this (L180).

264: it would be helpful to remind the reviewer that GPP here represents annual GPP.
We modified the text correspondingly.

297: this is far too many significant digits for a study of forest biomass.
The reviewer makes a confusion between the coma that separates the thousands with a dot. No decimal places are being used in the number. We removed the coma.

We also took the opportunity to reformat the Supplementary material and institutional addresses, in order to address the comments made by Daria Karpachova on the MS record.

On behalf of all the authors, respectfully,
Olivier Bouriaud